# A multi-method psychological autopsy study on youth suicides in the Netherlands in 2017: Feasibility, main outcomes, and recommendations

Saskia Mérelle[1]*, Diana Van Bergen[1,2], Milou Looijmans[1], Elias Balt[1], Sanne Rasing[3,4], Lieke van Domburgh[5,6], Maaike Nauta[7], Onno Sijperda[8], Wico Mulder[9], Renske Gilissen[1], Gerdien Franx[1], Daan Creemers[3,4], Arne Popma[6]

1 Research Department, 113 Suicide Prevention, Amsterdam, The Netherlands, 2 Faculty of Pedagogical and Educational Sciences, University of Groningen, Groningen, The Netherlands, 3 Child and Adolescent Psychiatry, GGZ Oost Brabant, Boekel, The Netherlands, 4 Radboud University, Nijmegen, The Netherlands, 5 Quality of Care & Innovation, Pluryn, Nijmegen, The Netherlands, 6 Child and Adolescent Psychiatry, Amsterdam UMC, Amsterdam, The Netherlands, 7 Department of Clinical Psychology and Experimental Psychopathology, University of Groningen, Groningen, The Netherlands, 8 Forensic department, GGD Noord- en Oost-Gelderland, Warnsveld, The Netherlands, 9 Youth healthcare, Dutch Centre for Youth Health (NCJ), Utrecht, The Netherlands

* s.merelle@113.nl

**Data Availability Statement:** Interview data cannot be shared publicly because of ethical restrictions: the dataset contains potentially identifying and

## Abstract

### Objectives

In the Netherlands, there was a sharp increase in the number of suicides among 10- to 19-year-olds in 2017. A multi-method psychological autopsy study (PA) was conducted to assess feasibility, identify related factors, and study the interplay of these factors to inform suicide prevention strategies.

### Methods

Coroners identified youth suicides in 2017 in their records and then general practitioners (GPs) contacted the parents of these youths. Over a period of 7 months, 66 qualitative interviews were held with the parents, peers, and teachers, providing information on precipitating factors and five topics involving 35 cases (17 boys and 18 girls, mean age 17 years). Furthermore, 43 parents and care professionals filled in questionnaires to examine risk and care–related factors. Qualitative and quantitative analyses were performed.

### Results

Although registration problems faced by coroners and resistance to contacting bereaved families by GPs hampered the recruitment, most parents highly appreciated being interviewed. Several adverse childhood experiences played a role at an individual level, such as (cyber) bullying, parental divorce, sexual abuse, as well as complex mental disorders, and previous suicide attempts. Two specific patterns stood out: (1) girls characterized by insecurity and a perfectionist attitude, who developed psychopathology and dropped out of school,

sensitive information and the Medical Research Ethics Committee (MREC) of Amsterdam UMC has imposed this restriction (registration number: 2018.651 – NL68348.029.18). Contact information: metc@vumc.nl; https://www.vumc.nl/research/overzicht/medisch-ethische-toetsingscommissie.htm.

**Funding:** The author(s) received no specific funding for this work.

**Competing interests:** NO authors have competing interests

and (2) boys with a developmental disorder, such as autism, who were transferred to special needs education and therefore felt rejected. In addition, adolescents with complex problems had difficulty finding appropriate formal care. Regarding potential new trends, contagion effects of social media use in a clinical setting and internet use for searching lethal methods were found.

## Conclusion

This first national PA study showed that, as expected, a variety of mostly complex clusters of problems played a role in youth suicides. An infrastructure is needed to continuously monitor, evaluate, and support families after each youth suicide and thereby improve prevention strategies.

## Introduction

Suicides account for 9% of all deaths among the youth, aged 15–29 years, and are a leading cause of death among adolescents worldwide [1]. In the United States and United Kingdom, an increasing trend in the suicide rate was found among 15- to 19-year-olds between 2010 and 2017, with young males having the highest suicide rate [2,3]. In the Netherlands, the suicide rate was stable among 10- to 19-year-olds (2.3 per 100,000) and relatively low compared to other countries (3.9 per 100,000) in previous years [1,4]. However, in July 2018, Statistics Netherlands (known as *Centraal Bureau voor de Statistiek* in Dutch) reported that the suicide rate was 1.7 times higher (4.0 per 100,000) in 2017 than in 2016 (2.6 per 100,000) [5]. As a result, the Ministry of Health directed a research group under the auspices of 113 Suicide Prevention, Amsterdam, the Netherlands, to examine the background on youth suicides in 2017, with the aim of recommending suicide prevention strategies on short notice.

Regarding the aetiological factors for youth suicides, the most described socio-demographical risk factors include the male gender, late adolescence (16–18 years) period, living alone, a family history of suicide, and parental mental health problems [1,6,7]. Mental health problems such as previous suicide attempts, psychotic symptoms, eating disorders, depression, substance use disorders, personality disorders, and post-traumatic stress disorder (PTSD) are also important risk factors [6,8,9]. Furthermore, some environmental factors have strong empirical evidence, such as early childhood adversities (e.g. sexual abuse, emotional neglect, and bullying), whereas other factors yield mixed evidence, such as contagion–imitation effects by peers and media influences [1,9,10]. Finally, general psychosocial factors such as low self-esteem, hopelessness, impulsivity, anger, aggression, and loneliness are moderately to strongly related to suicidal behaviours [1]. However, all of these factors not only are related to suicidal ideation and attempts but can also be regarded as general risk factors for developing psychopathology. Moreover, youth suicides are infrequent, and direct relations with risk factors are difficult to establish [6,11].

Despite the body of evidence regarding general risk factors for suicidal behaviour, little is known about how the complex interplay of these factors contributes to youth suicides [9]. Psychosocial stressors may play a role in triggering the process directly preceding youth suicides. The triggers include interpersonal losses (e.g. relationship break-ups), school-related problems and academic stress, acute conflicts with peers or parents, traumatic events (e.g. cyberbullying), and disciplinary problems (e.g. police contacts) [9]. It is, therefore, important to identify

the patterns of factors underlying youth suicides and the role of triggers in the onset and course of suicidal behaviour. In addition, there is a paucity of research on the role of sexual orientation and gender identity in youths who died by suicide [12,13].

A further concern is the time spent on social media and online gaming, which has been linked to the deterioration of the mental health of adolescents over the last decade [2,14]. Although there are many benefits of social media use, it was found to be attractive for individuals with suicidal tendencies to express their distress and interact with similar others in an anonymous and easily accessible environment [15–17]. Online media use is potentially harmful when normalizing suicidal behaviour and triggering effects (i.e. contagion), encouraging competitive behaviour, and providing information on suicide methods [18]. In addition, a case report study highlighted the possible contagion effects of participation in the online "Blue Whale Challenge", where predominantly female players were imposed to repeatedly cut themselves [19]. Furthermore, victims of cyberbullying are almost three times more likely to attempt suicide and two times more likely to have suicidal ideation compared to non-victims [20,21]. Sex differences also seem to play a role because frequent social media use was found to be associated with higher depressive symptoms and suicide-related outcomes in young females, whereas problematic video gaming was associated with depression and suicidal ideation in young males [14,22]. Another recent trend is the potential influence of online series, such as the release of the Netflix show *13 Reasons Why* (13RW) in 2017. The fictional portrayal of the suicide of a young girl in this series might have caused imitation effects, and it was found to be associated with an increased number of youth suicides in the United States and Canada [23,24].

In a recent study, we examined demographical factors, methods, and seasonality as a first step towards investigating suicides among 10- to 19-year-olds in the Netherlands in 2017 [25]. As expected, more adolescent Dutch boys (62%) died by suicide in 2017 compared to Dutch girls (38%), and a higher number of old Dutch teenagers (15–19 years; 86%) died in that year compared to young Dutch teenagers (10–14 years; 14%) [25]. The method most frequently used was strangulation or suffocation (60%), followed by jumping or standing in front of an incoming train (28%), which indicates that no new methods were used [7,25]. The monthly frequency of suicides was equally spread throughout 2017, with a small peak in July and October. There were also regional differences, with 2 out of 12 provinces showing a relatively high suicide rate (Noord-Brabant, 6.8 per 100,000; Gelderland, 5.1 per 100,000; and the Netherlands, 4.0 per 100,000), indicating that point clusters (suicides localized in both time and place) might have taken place [26]. In addition, the proportion of youths with a migration background (31%) was higher than that of the youth population as a whole (25%) [27].

In summary, considerable evidence exists for general risk factors for youth suicides. However, the aforementioned studies lack insights into factors that played a role at the individual level, and little is known about the interplay of these factors [9]. Moreover, research is urgently needed to investigate potential new trends underlying youth suicides [2]. Youth suicides seriously affect families, relatives, and communities, but they are a statistically rare event [28]. It is, therefore, important to continuously monitor and evaluate youth suicides in order to learn from fatal cases. Although psychological autopsy (PA) is considered a valuable instrument to gain a better understanding of suicides, it has never been applied in the Netherlands [29]. Usually, face-to-face structured or semi-structured interviews are conducted with relatives of the deceased, and sometimes close friends, psychologists, or doctors are also included as respondents [29]. When implementing PA in the Netherlands, feasibility questions need to be answered, such as which infrastructure and multiple data sources are currently available and how willing are suicide descendants and professionals to participate?

### The present study

This study aimed to, first, identify factors related to youth suicides and examine the interplay of these factors to inform suicide prevention strategies in health care and community settings [30]. The second aim was to investigate the feasibility of applying PA in the Netherlands. To our knowledge, this is one of the first PA studies using a multi-method approach with an open, narrative start, and semi-structured components of predefined topics were supplemented by questionnaires. It was not possible in the available time to conduct a case–control study, and suicide descendants from only 2017 were included.

We investigated several topics on the basis of previous literature and statistical findings and current potential trends in suicide among the youth. First, the developmental stage of adolescence was examined broadly, particularly the transition from early to late adolescence. In this article, remarkable findings related to youth suicides are presented (e.g. school career, social relationships, family composition, and substance use) [31]. Second, key issues around youth health care were studied, such as psychiatric diagnosis, treatment, and perceived quality of care [32]. Third, suicide clusters and imitation effects through social media were studied, including online challenges, games, and series [18,23,26]. Fourth, related factors in the lives of sexual and gender minority youth were looked into [33,34]. Fifth, the role of cultural and migration factors was studied in suicidality of the youth with a migration background [35]. Last, psychosocial stressors and behavioural changes during the last period before death by suicide occurred were examined [9].

## Materials and methods

### Ethics

The Medical Research Ethics Committee (MREC) of Amsterdam UMC approved the study (registration number: 2018.651 –NL68348.029.18). All participants gave their written informed consent. The study design for qualitative components followed the consolidated criteria for reporting qualitative research (COREQ) [36].

### Study design and procedure

A multi-method study was conducted. In this case, qualitative data and survey responses were applied to identify related factors and emergent patterns of youth suicides in the Netherlands. Emphasis was placed on qualitative dimensions by delving into the meanings and perceptions of parents and other informants. We used questionnaires to quantify the frequencies of established risk factors and care-related factors to supplement the statistical findings on socio-demographic characteristics [7].

Data were collected from February to October 2019. The recruitment of participants was carried out in phases: (1) Coroners were asked to search their records to select 10- to 19-year-old adolescents who died by suicide in their regions in 2017, together with the contact details of general practitioners (GPs). (2) Coroners sent letters to GPs asking them to inform the parents of those who died by suicide about the research. (3) The parents who were willing to participate were contacted by an interview coordinator, who further explained the research being conducted, scheduled the interviews, and coordinated aftercare if needed. (4) The parents were asked to approach three other informants (peers or siblings, teachers or employers, and health care professionals) who knew their child very well and provided the contact details of those willing to participate [37]. In addition, the parents were recruited through social media, such as via calls in local newspapers or websites of professional associations. This was done carefully, adhering to media guidelines to avoid a copycat effect among adolescents

and respect the feelings of the bereaved families [38]. All the parents, peers, and siblings who were willing to participate were screened by the interview coordinator and excluded from the study if they showed a high risk of suicidal behaviour (score ≥21 on the Suicidal Ideation Attributes Scale) [39]. The parents who received inpatient care in a mental health institution were also excluded from the study. In addition, GPs were informed about study participation.

The parents participating in the study were interviewed one-time for 2–3 hours in their homes, and they received a digital questionnaire by email afterwards, which took approximately 30 minutes to complete. A senior interviewer (i.e. an experienced care professional), together with a junior researcher, interviewed the parents and made audio recordings. One week after the interview, the senior interviewer made calls to check up on all the parents to confirm their well-being. Siblings and friends were interviewed by the senior interviewer. The interview lasted approximately 1.5 hours. Minors (<16 years of age) were interviewed in the presence of their parents, whereas peers (≥16 years of age) could bring a support figure to the interview. Teachers or employers were interviewed for 1.5 hours by the senior interviewer. Health care professionals received two digital questionnaires, followed by a 30-minute telephonic interview with the senior interviewer to discuss the completed questionnaires. All informants were allowed to see the verbatim transcript of their interview.

It was expected that around 60% of the parents would be willing to participate because the response rate in previous PA studies varied between 44% and 66% [8,40–42]. While setting up this study, we aimed for a minimum of 10–12 cases per subgroup (topic) to reach saturation [43]. In qualitative research, this refers to the moment when the key components and patterns for the subgroup in the data are visible. In this study, these pre-identified subgroups concerned younger versus older adolescents; adolescents who had received youth (mental) health care versus those who had not; sexual and gender minorities versus heterosexually identified, adolescents who were active social media users versus passive users; and adolescents with a migration background versus youths with a native background.

## Measures

The interview was based on previous PA studies in Ireland, Belgium, Norway, and the United Kingdom and supplemented with questions from experts in the research group and advisory committee [8,41,42,44]. This led to five topics that seemed to bear relevance for the youth today. The interview started with an open, narrative component, where parents, peers, siblings, or teachers responded to a broad open question about what had induced the development of suicidal behaviour in the young person [44]. Next, interviewers asked about critical events and behaviours in the last period before death by suicide. Subsequently, semi-structured questions about pre-identified topics were asked (adolescence, youth health care, clusters and social media, sexual and gender diversity, and ethno-cultural and migration factors). Throughout the interview, interviewers used narrative-pointed questions aimed at letting the respondents share stories about the life, emotions, thoughts, and behaviour of the adolescents [45]. All the interviews were transcribed verbatim by data entry professionals who signed a confidentiality clause. The interview guide was tested first among four cases (youth who died by suicide in 2018), and a small number of minor adaptations to the instrument were made accordingly.

The questionnaire consisted of a selection of self-report questions and scales used in cross-sectional surveys carried out by Municipal Healthcare Services, the Netherlands, to monitor health status and lifestyle behaviours among young inhabitants [46]. For the purpose of this study, the questions were adjusted to a format for parent reports. The parents also completed the parent form of the validated Strengths and Difficulties Questionnaire (SDQ), a 25-item diagnostic tool to identify children at high risk for emotional and behavioural problems [47].

The cut-off points from Municipal Healthcare Services were used to categorize scores on the psychosocial, emotional, and behavioural scales into "borderline" and "elevated" scores [48]. The history of suicidal behaviour was measured with several validated items on suicide and self-harm, supplemented with questions from experts on childhood life events and trauma [49]. The questionnaire for health care professionals was based on the Irish PA study and was adjusted to Dutch youth health care [8].

## Qualitative and statistical analyses

First, the feasibility of the PA procedure was evaluated using experiences of the research team, participants, and professionals involved, which revealed several benefits and pitfalls. Second, descriptive quantitative analyses were performed using SPSS (v.25.0) to analyse the survey responses and calculate the mean age and prevalence of socio-demographical and pre-identified risk factors. For privacy reasons, numbers smaller than 5 are not presented in Table 1. Third, the constant comparative method was used to analyse the interviews aided with ATLAS.ti [50]. Several steps were taken for the qualitative analyses: (1) Both interviewers wrote a summary of each interview immediately after the interview, independently of each other, focusing on key events and relevant factors in the life and events related to the suicide of adolescents. (2) The coding team consisted of 6 researchers, three researchers started with inductive coding of the interviews of the first three parents, where they coded all fragments with relevant information on the main subject and created a code list. (3) All 6 researchers suggested changes or supplemented this code list through discussion, as well as added useful codes derived deductively from the interview instrument, when they were approved by the senior researcher. (4) All coded transcripts were subjected to examination by a second coder, who checked, supplemented, or changed the codes of each transcript to increase the reliability of the coding. (5) Four researchers extracted the output of the coded fragments per study topic and summarized the results. (6) A researcher not involved in making the initial summaries checked these summaries for comprehensiveness. (7) Five researchers made an axial comparison to detect patterns across suicide cases regarding pre-identified topics. Finally, the main findings are presented in this article, and more detailed results of the qualitative analyses are presented in subsequent articles.

## Results

### Study population

As provided in Table 1, a total of 95 respondents took part in the study, who were related to 35 adolescents who died by suicide in 2017. The parents of four other adolescents were willing to participate, of which three parents withdrew from the research during the inclusion procedure (participation burden) and one parent withdrew after the interview (and refused audio recording). There were 12 refusals from parents and 8 on behalf of parents, for example, in case of complicated grief indicated by GPs (see the section "Experiences regarding feasibility"). The researchers were not able to reach the parents of 22 suicide cases because of external factors; for example, the parents had moved or the GPs were unknown. Consequently, the response rate was 59% (the parents from 35 of 59 cases who were reached).

In addition, no respondents were excluded from the study on the basis of scores on suicidal ideation or mental health institution stays. Furthermore, a total of 66 qualitative interviews were conducted over a period of 7 months: 37 face-to-face interviews with parents (in two cases, the parents were interviewed separately) and 29 face-to-face interviews with peers, teachers, and an employer (see Table 1). In addition, 11 telephonic interviews were conducted with health care professionals.

**Table 1. Number of respondents and characteristics adolescents in study population.**

| | Respondents (N = 95) |
|---|---|
| **Interviews** | |
| Face-to-face interviews parents, n = 37 | 54 |
| Face-to-face interviews peers, siblings, n = 18 | 19 |
| Face-to-face interviews teachers, employer, n = 11 | 11 |
| Telephonic interviews healthcare professionals, n = 11 | 11 |
| | **Adolescents (N = 35)[a,b]** |
| **Gender, n (%)** | |
| Boys | 17 (49) |
| Girls | 18 (51) |
| **Age, mean (SD)** | 16.9 (1.5) |
| **Ethnicity, n (%)** | |
| Native Dutch | 31 (89) |
| Non-native/Western | * |
| Non-native/non-Western | * |
| **Region, n (%)** | |
| Noord-Brabant | 8 (23) |
| Gelderland | 7 (20) |
| Zuid-Holland | 7 (20) |
| Other | 13 (37) |
| **Suicide method, n (%)** | |
| Strangulation or suffocation | 20 (57) |
| Jumping or standing for a train | 12 (23) |
| Other | * |
| | **Adolescents (N = 32)[c]** |
| **Education, work, benefit, n (%)** | |
| HAVO/VWO (higher secondary education) | 14 (44) |
| MBO (vocational education) | 8 (25) |
| VMBO (lower secondary education) | 5 (16) |
| No school (work, benefit, clinic) | 5 (16) |
| **Family composition, n (%)** | |
| Both parents | 15 (47) |
| Single parents (mother) | 6 (19) |
| Other (co-residence, living alone, other people, mother and new partner, single parent father) | 11 (34) |
| **Divorced parents, n (%)** | |
| Parents are divorced | 15 (47) |
| **Religion, n (%)** | |
| Belongs to religion | 9 (28) |

[a] Age, ethnicity and region comes from the interviews.

[b] Due to privacy, numbers smaller than 5 are not presented.

[c] Three parents did not complete the questionnaire, these results concern 14 boys, 18 girls.

**Characteristics adolescents.** Table 1 presents the characteristics of adolescents who died by suicide in 2017 and were included in this study. The group consisted of 17 boys and 18 girls, with a mean age of 16.9 years (*SD* = 1.5). All children were born in the Netherlands; two had a parent or both parents who were born in a foreign Western country; and two had a

parent or both parents who were born in a non-Western country. Considering these small numbers, reporting on the relevance of ethno-cultural factors on an aggravated level was not possible, and thus, we omitted this topic. Because none of the participants was transgender, we only referred to youths who were lesbian, gay, or bisexual (LGB). The topic clusters included three adolescents, and consequently, we briefly describe these findings in the section "Potential contagion effects and social media use". By contrast, the subgroups within the topics adolescence, youth health care, and social media were large enough to achieve saturation: 23 old teenagers (17–19 years), 12 young teenagers (14–16 years), 22 received youth health care treatment at the moment of death, 13 did not receive treatment, 20 were active social media users; and 10 were LGB or questioning one's sexual orientation.

## Experiences regarding feasibility

A number of bottlenecks appeared during the recruitment procedure. First, the Netherlands is divided into 25 regions of Municipal Healthcare Services. Usually, the regional coordinator referred the researchers to a local coroner, who often had difficulties finding the number of youth suicides that the regional coordinator had indicated. Only two coroners could efficiently identify youth suicides in their regions in 2017. Furthermore, in some regions, coroners did not automatically register the GP or the GP was unknown when they established the cause of death. It also appeared difficult for some coroners to find the GP retrospectively. Consequently, many GPs were missing in their registrations. Because of an approved MREC amendment, parents could directly receive research letters from coroners if GPs were missing. Furthermore, one coroner who worked in a region with a high suicide rate among the youth refused to participate because of privacy reasons. Second, five GPs and one coroner decided not to inform the parents about the study because they thought that the parents were too vulnerable because of complicated grief. In addition, two GPs had contacted the parents a long time ago and were hesitant to contact them again. A professor in suicide prevention contacted these GPs to give them advice about difficult cases. Third, it was difficult to recruit other informants through parents because 13 parents indicated that they could not or would not involve other informants, and informants who were eligible according to the parents often did not respond to emails or calls from the interview coordinator.

Regarding the benefits of the PA procedure, most parents appreciated being interviewed, specifically the ability to share everything about their children's lives with interviewers who were understanding and perceived as non-judgemental. Of all relatives, five parents, one peer, and one sibling were advised to contact their GPs; one parent was referred to a therapist specialized in complicated grief; and eight parents already received help. The facilitation of bereavement support, therefore, seemed an important part of the PA procedure.

## The developmental period of adolescence

The interviews revealed, in a first pattern, that seven adolescents—mainly girls in the highest academic tracks in secondary education—increasingly suffered from perfectionism and strived after the highest possible grades in school. The tendency to be a perfectionist had mostly started in their early adolescence. The fear of academic failure and pressure to perform better all the time led to increasing feelings of insecurity, especially about getting schoolwork done on time. For some of them, this negative spiral was reinforced by absenteeism in school because of eating problems and/or psychological and psychosomatic complaints or because of spending time as an inpatient in a clinic. A second pattern consisted of seven adolescents, mostly boys, who had a diagnosis—such as dyslexia, attention-deficit/hyperactivity disorder (ADHD), or autism—which led to stagnation in their educational progress, although their

parents generally thought that they did not lack the cognitive abilities to perform well. Negative school experiences affected their already-low self-esteem, where demotion to another type of education and the fear of ending up with bullies and rascals in special needs education were sometimes triggers in the last period before death by suicide.

Apart from these two patterns, several adverse childhood experiences were reported by the respondents. According to the 32 questionnaires that the parents filled in, 15 (47%) youths had divorced parents, 13 (41%) ran away from home at least once, 9 (28%) came in contact with the police, and 9 had financial problems (28%). In addition, 4 (13%) suffered from sexual abuse and 8 (25%) physical abuse. The interviews showed that, in many cases, there were recurrent situations of high tensions in the family home. In 10 cases, problematic divorces put pressure on contact with their parents, mostly fathers. It seemed that some youths felt emotionally neglected by their fathers, or sometimes, they did not get along well with the new partner of one of the parents. In six cases, domestic violence was mentioned, and sometimes the police were involved. It is worth noting that, in three cases, youths were said to have acted violently towards their parents. The interviews also revealed that (long-term) sexual abuse, date rape, and rape by a stranger or family member—as well as online sexual harassment—had a negative impact on the lives of four adolescents. Three youths had developed PTSD as a result of the abuse.

The questionnaires showed that 15 (47%) adolescents had experienced bullying and 7 (22%) were bullied via the internet. According to many parents and peer respondents in the interviews, being the target of harassment and bullying had caused permanent harm to their children, making them subject to increased feelings of sadness and anxiousness and a lack of belongingness. Another common finding was that many youths were insecure about their friendships and sensitive to perceiving rejection, whereas their peers thought they were sociable and likeable.

The parents responded in the questionnaires that 11 (34%) adolescents sometimes used soft drugs and 5 (16%) hard drugs. The parents did not know about their children's drug use in three to five cases. In addition, 7 (54% of the group who sometimes drank alcohol) were heavy drinkers. The parents reported in the interviews that drug use was directly related to suicidal behaviour in five cases. They saw that the mental well-being of their children deteriorated as a result of drugs or alcohol use. In two cases, drug use led to a psychotic episode and coincided with suicide.

## Youth (mental) health care

As reported in the questionnaires by the parents of 32 adolescents, 23 (72%) suffered from suicidal ideation, 17 (53%) made at least one suicide attempt, 17 (53%) had a family member with a mental disorder, and 4 (13%) had a parent or sibling with a severe psychiatric disease. As reported by the parents in the 31 SDQs, 11 (36%) adolescents had borderline scores on psychosocial problems in the 6 months prior to their death and 15 (48%) had elevated scores on psychosocial problems. In addition, 20 (65%) had moderate or elevated scores on emotional problems and 12 (39%) scored moderately or elevated on behavioural problems.

At the time of their death, 22 (63%) of the 35 adolescents were enrolled in youth (mental) health care, health care, or youth care (i.e. general practice–based nurse specialists in primary care, inpatient and outpatient youth health care, child protection, addiction care, forensic care, and trauma care). Of the 13 adolescents who did not receive treatment around the moment of death, 7 had never received professional help, 4 had received treatment earlier in life, and 2 had only one intake interview or consultation with a GP.

Six of the 22 adolescents were admitted to a clinic at the time of death, of whom 5 stayed in a closed ward facility. In addition, one adolescent died in an assisted living residence, one during her stay with a foster family, and one during hospital admission (after a suicide attempt while having psychotic symptoms). The six adolescents who were admitted to an inpatient clinic were able to get access to means, nevertheless, by which they could strangle or suffocate themselves. They were sometimes left alone in their rooms, usually against what had been agreed upon or through miscommunication. Moreover, another youngster was discharged from the hospital after a suicide attempt without further assistance or follow-up and then died by suicide.

Seventeen (49%) adolescents had a psychiatric diagnosis. Eleven had combinations of psychiatric diagnoses, such as depressive disorder, eating disorder, anxiety disorder, attachment disorder, bipolar disorder, PTSD, autism spectrum disorder (ASD), oppositional defiant disorder (ODD), mild intellectual disability, alcohol abuse, psychotic problems, and traits of personality disorders (borderline, avoidant, or narcissistic), with comorbidity ranging from two to nine diagnoses per young person. Six youths had combinations of developmental disorders, such as ADHD, attention deficit disorder (ADD), pervasive development disorder not otherwise specified (PDD-NOS), dyslexia, PTSD, antisocial personality disorder, and addiction. Fourteen of these 17 adolescents with diagnoses had complex problems, with a great deal of comorbidity requiring specialist treatment. In addition, five adolescents had not been diagnosed yet, three of whom were believed to suffer from borderline personality traits.

For nearly all the youth enrolled in care, the treatment focused on crisis care—reducing psychiatric symptoms such as depression—or on how to deal with their diagnoses. Seven out of 11 care professionals reported that the diagnosis and treatment also focused on suicidal behaviour, which consisted of personalized safety plans aimed at reducing suicidal behaviour, conducting regular assessments of suicide risk, preventing suicide by admission to a closed care facility, using the family system for safety agreements, and providing individual therapy sessions to discuss suicidality. In addition, most adolescents had seen their health care professional in the last days or weeks before their death.

As mentioned previously, many adolescents suffered from multiple and complex psychiatric disorders. In the Netherlands, there are relatively few places with expertise for the treatment of multiple psychiatric problems, and therefore, long waiting lists exist for these facilities. As a result, these adolescents had to stay at home for a long time before getting the specialized care they needed, which seemed to worsen their problems. The lack of continuity of care, particularly changes in the therapists assigned and institutions chosen, was an important bottleneck in the care trajectory according to both parents and professionals. The aggravation of the mental health of adolescents was also a result of feeling a burden to their caregivers, as well as feeling hopelessness and unwanted.

Another issue was the compulsory transition from youth to adult psychiatry. In the Dutch care system, this transition takes place when adolescents turn 18 years old. This transition made adolescents anxious and insecure about their future. Furthermore, the parents felt restricted by Dutch legislation, which rendered it more difficult for parents to have a say in and access to treatment information of their children when they were 16 years or older. They sometimes felt that professionals did not have a complete or realistic view of the mental health of their children or that confidentiality clauses were more important to professionals than the well-being of their children. For example, the parents recalled situations where therapists did not listen to them when they tried to explain the severity of their children's situation in relation to suicide risk.

Finally, six adolescents indicated that they wanted to die with the help of the expert centre of euthanasia that exists in the Netherlands. The parents explained how, in these cases,

children had given up any hope of improvement and wanted to know how to die painlessly, easily, or neatly. Two adolescents actually contacted the expert centre of euthanasia, but the centre did not comply with their requests.

## Potential contagion effects and social media use

Two regions (Noord-Brabant and Gelderland) showed a relatively high suicide rate in 2017. The parents in Noord-Brabant reported a cluster of four suicides in one day at the same location. Other parents in Noord-Brabant reported a cluster of four students who ended their lives within a single school year, of whom two were classmates. There were also indications for a cluster in Noord-Holland, where the parents reported that two adolescents died at a close distance of the location where their children died by suicide. The parents often did not know whether there had been (online) contact among the suicide victims or their children participated on internet forums with other suicidal adolescents. Some parents had stopped monitoring their children's social media activities after a certain age and were confident that their children used social media responsibly. In 15 cases, the parents reported that the police had checked the telephones and computers of their children after death by suicide. They said that the police did not find anything remarkable; however, some of them indicated that their children were smart enough to delete any digital trace.

As for potential contagion effects, the parents reported in the questionnaires of 32 cases that 16 (50%) knew someone with suicidal ideation and/or suicide attempts and 9 (28%) knew someone who died by suicide. Deaths by suicide in the direct environment concerned family members, classmates, fellow patients, or a parent of the best friend. The suicides of two rock artists were also mentioned. In addition, the parents often mentioned that their children had deliberately chosen a certain location to die (e.g. a remote place next to the track or a location that they knew from previous train suicides). Furthermore, adolescents could easily search for and find information online about lethal suicide methods and were able to order them online. Using the internet, some adolescents were able to prepare their suicide quietly.

Social media use played an important role in the daily lives of 20 adolescents, and a variety of impacts on suicidal behaviour were observed. Five girls had a second social media account that their parents did not know of before their death or had access to. On this "secret" account, they shared photos of their own self-harm and suicide attempts, depressive memes, and poems with their peers. In this respect, one peer stated that social media seemed to reinforce an "online suicidal identity". Her friend followed everything about depression via Instagram; the sick role was "who she was" and social media helped hold on to that suicidal identity. In addition, one parent indicated the problem of algorithms in social media, which kept on feeding their child with negativity. Another parent mentioned that cutting seemed to be a hype on Instagram, which encouraged her daughter to self-harm in primary school.

In about 10 cases (9 females and 1 male), respondents described how important social media was for suicide victims during their admission to a clinic. They needed social media to communicate with the outside world and frequently used WhatsApp to contact fellow patients. They were able to build friendships with fellow patients and follow their recovery stories; however, exchanging self-harm experiences led to competitive self-harming behaviour. In addition, adolescents who knew each other from the clinic started caring for each other via social media, which parents considered burdensome for their children. One parent stated that her daughter had become a "care professional" via WhatsApp rather than the person who needed help.

The informants did not indicate a direct link between problematic gaming and suicidal behaviour, but they did point out that mental health problems—such as sleep problems, concentration problems at school, and becoming increasingly isolated—were triggered after

intense gaming. Furthermore, challenges could be related to three suicide cases. In one case, there were indicators that the youth participated in the Blue Whale Challenge, whereas the other challenge concerned an app that stimulated self-harming behaviour. In a third case, the suicide resembled a choking challenge, but according to parents, it was probably an accident.

The Netflix show 13RW was first shown on 31 March 2017. In seven interviews, parents indicated that their children had seen the show. Five adolescents found the episodes impressive and exciting; however, the parents did not discuss the show extensively with their children. In addition, the adolescents with long-term mental health complaints either avoided the show or were not affected by it because they had already experienced far more serious things in their lives. No further evidence was found in the interviews that online series were directly related to suicidal behaviour, except for one online series that seemed to have triggered a copycat train suicide.

## Youth who were LGB

Five adolescents (15%) had a homosexual, lesbian, or bisexual orientation according to their parents, and in four cases, the parents reported that their children's sexual orientation was "unknown". The interviews showed that 10 adolescents had discussed with their parents that they felt attracted to the same sex or wondered if they were. According to the parents, their children's search for their sexual identities did not play an important role in the process leading to suicide. The other respondents (the mentor and friends) sometimes had a different opinion in this matter, as they mentioned coming-out stress or rejection by family or peers and bullying. For example, four young people were not (fully) believed by their parents or peers when they said that they were attracted to the same sex. The parents believed it was either "copycat" or "seeking attention" behaviour or thought they were too young to be certain or knowledgeable about their sexual identities. For some of them, there was a critical event in the last months before death by suicide related to LGB issues, such as sexual abuse or being bullied, which had taken place shortly after they had come out.

## Psychosocial stressors and behavioural changes in the last period

Regarding psychosocial stressors in the last months prior to death by suicide, an event or change in youth health care led to a deterioration of the well-being of 12 adolescents—for example, a transfer from youth to adult care, an out-of-home placement by youth care, or being rejected for a specific treatment. For six adolescents, school-related problems played a role, such as failing an examination, an intake at a school for special needs education, and receiving low grades in high school (blocking future study ambitions). Eleven adolescents experienced a setback in their peer or family relations, such as disappointments in romantic relationships and feeling like a burden to their parents because of their poor mental health. In addition, five youths worried about the responsibilities of becoming adults, such as living independently.

Regarding behavioural changes just before their suicide, some adolescents withdrew from their families and social contacts in the last weeks. They did not stick to agreements, stayed away from school, spent more time alone in their bedrooms, or deleted their social network accounts. In addition, some adolescents had less energy, had sleep problems, and were more stressed or angry, and a few youths seemed to sedate themselves by substance use. Psychotic complaints played a role in four adolescents.

Among adolescents who had already attempted suicide, their suicidal behaviour (e.g. self-cutting) became more severe in the last period, and some started to talk more about their suicidality. By contrast, in almost half of all cases, the parents indicated that their children seemed

to be getting better during the period preceding their suicide and they enjoyed nice moments with them. The parents, therefore, thought that suicidality or other mental problems had decreased or disappeared. In retrospect, these moments were apparently farewell moments to their loved ones. Some adolescents also sent messages to loved ones just before their deaths or tagged dear friends or family members in loving messages. Finally, 11 adolescents—both religious and non-religious—expressed that they assumed they would enjoy afterlife, which seemed like a reassuring idea.

## Discussion

The primary aim of this study was to gain insights into youth suicides by addressing related factors and the interplay of these factors to inform suicide prevention strategies in health care and community settings in the Netherlands. In our multi-method PA study, 54 parents and 41 other informants were interviewed over a period of 7 months, providing information on a total of 35 adolescents (14–20 years old) who died by suicide in 2017. The interviews focused on the interpretations respondents provided of the adolescents' history of suicidal behaviour, last period before death by suicide occurred, and pre-identified topics related to adolescence, youth health care, contagion effects and social media, sexual orientation, and gender identity. The secondary aim was to evaluate the feasibility of this first PA study in the Netherlands. Overall, the parents appreciated the qualitative approach in which they narrated the life course of their children. The registration problems faced by coroners and resistance to contacting bereaved families by GPs hampered the recruitment procedure.

As expected, several adverse childhood experiences played a role at an individual level, such as (cyber)bullying, parental divorce, sexual abuse, as well as complex mental disorders, and previous suicide attempts. Our findings showed two specific patterns from the beginning of secondary school that gave, as far as we know, new insights into youth suicides. This study further revealed several bottlenecks regarding procedures in youth (mental) health care trajectories. Regarding potential new trends underlying youth suicides, we found contagion effects of social media use in a clinical setting and internet use for searching lethal methods. Moreover, this is one of the first PA studies showing that minority stressors (homophobic bullying and coming-out stress) played a role in the lives of some youths who were LGB. Contrary to previous studies, we did not find any evidence of a relationship with the release of 13RW in 2017. In addition, only three indications of point clusters were found, and online challenges played a role in only two cases.

The first pattern emerged during early adolescence and concerned insecure girls with a perfectionist attitude, who developed psychopathology and dropped out of the regular schooling system. The second pattern concerned boys with learning disabilities and/or developmental disorders, such as dyslexia, autism, or ADHD, who were transferred to special needs education and therefore felt rejected. A recent study showed that Dutch secondary school children experienced significantly more pressure because of schoolwork in 2017 compared to previous years (2001–2013) [51]. This upward trend was particularly observed from 2013 (28%) to 2017 (35%), and girls showed the highest increase (9%) compared to boys (6%). It seems likely that certain personality traits made some adolescents more vulnerable to performance pressure. Perfectionism and neuroticism are two of the personality traits that have been identified as predisposing factors for suicidality [52].

Finding appropriate care was difficult for adolescents with complex mental disorders. This study highlighted several issues in care trajectories, which varied from waiting lists, lack of continuity of care, compulsory transition from youth to adult care at age 18 years to a low sense of involvement on the part of parents, which was particularly due to legislation

constraints when their children became 16 years or older. In addition, one-third of the youth experienced critical events in youth health care in the last months before their death that seemed to worsen their conditions. However, when interpreting these results, we should note that more than 400,000 adolescents received youth health care in the Netherlands in 2017, whereas death by suicide occurred only in a small group [53]. Our findings also underline the importance of actively involving the family system in the treatment, using personalized safety plans and arranging aftercare according to the needs of adolescents, given the fact that risk assessment by clinicians is often too inaccurate to be clinically useful [54].

Our social media results are mostly in line with previous studies. A previous study analysed posts on Twitter, Tumblr, and Instagram, using the hashtag #cutting, and found that 60% of the sampled posts showed graphic content of (active) self-harm (particularly on Instagram), nearly 50% contained negative self-evaluations, and only 10% discouraged self-harm [55]. Likewise, a study using hashtags #suicide and #suicidal found that Instagram was frequently used for suicide-related communications [56]. Our results add to these findings by demonstrating that exposure to suicidal content through social media was contagious for adolescents who eventually died by suicide. We also found that some adolescent girls had a hidden social media account where they shared content of deliberate self-harm. In addition, it appeared that they used social media to create an online suicidal identity that refrained from looking for positive content and pursuing future-oriented plans. We replicated the finding that social media can have both positive and negative effects in our study linked to the subgroup of clinical patients [17]. Social media was beneficial for 10 adolescents while being admitted to a clinic, as they received peer support, shared recovery stories, and felt needed when they offered care for another youth via social media. However, the exchange of self-harm messages via social media was considered harmful and encouraged a mutual competition among young patients.

Regarding suicide clusters, we could only examine 35 of the 81 teenage suicides in 2017 and were not allowed to use the data of Statistics Netherlands at the individual level because of privacy reasons. Therefore, a national suicide information system is needed for real-time monitoring of suicide clusters among youths. If real-time data are available, preventive actions can be taken at an early stage. Contrary to previous studies, no evidence was found for "copycat suicides" of 13RW in 2017. The small number of suicides and using parents as important sources of information might have limited a comprehensive study of the impact of 13RW. However, our results indicate that neither identification with the main character nor imitation of suicidal behaviour took place in the study group [23]. By contrast, hanging and train suicide were the most common suicide methods in Dutch youths in 2017. According to many respondents, adolescents knew about lethal methods because they had insider information of previous suicides or had searched the internet for techniques.

To our knowledge, this is one of the first studies that examined the relationship between sexual orientation and suicidality in youths who died by suicide. This study revealed that a relatively high proportion in our sample felt attracted to the same sex. Our findings further provide some evidence that being LGB or questioning one's sexual orientation is likely to increase psychosocial stress because of minority stressors, which in turn increases suicide risk.

The finding that psychosocial stressors played a triggering role (e.g. transitions in health care, school-related problems, relationship break-ups, and feelings of burdensomeness towards their parents) is in line with previous research [9,57]. Although adverse events are seldom a sufficient cause for suicide, their importance lies in their function as precipitating factors in youths at risk for suicide [52]. In this respect, we observed several behavioural changes that occurred just before suicide (e.g. withdrawing from social contacts, deleting social media accounts, being talkative and cheerful, having sleep problems, and indulging in more self-harm acts). Remarkably, one-third of the adolescents, both religious and non-religious,

seemed reassured in the last period because they believed in afterlife. Although religion often acts as a protective factor in suicide, this finding needs elaboration in future research, particularly as a promising aspect for intervention [58].

## Strengths and limitations

This study has distinct strengths. First, we used a predominantly qualitative multi-method approach that was valued by the participants and offered insights into the mechanisms underlying youth suicides. This is important in understanding a complex phenomenon, such as youth suicides, in which multiple risk factors are known to be involved [11]. We used (validated) questionnaires to ensure data collection in a short time frame and supplemented the findings of Statistics Netherlands. In some cases, the survey responses resulted in new information (e.g. several youths had run away from home). The overlap in the aims and objectives of the interviews and questionnaires should be reduced in the next research, making the PA procedure more efficient. Second, we used a large research team and an advisory committee with various experts, which enabled us to complete a large-scale PA study within 1.5 years. Third, we focused on current trends, such as social media, and unexplored topics in the PA procedure, including sexual orientation and gender diversity. These results may be fruitful in preventing contagion effects through social media and in developing interventions that may reduce minority stressors in youths who are LGB.

This study has some limitations: First, the parents asked other informants for participation. This procedure might have increased the likelihood that shared perceptions and information bias would occur. However, the interviews showed that peers mostly provided additional information, for instance, on specific topics such as social media and sexual orientation. The recruitment of teachers and care professionals took more effort and lagged behind the strict time schedule. Hence, more survey data from care professionals might have provided more insights into the degree of adherence to the multidisciplinary clinical guideline for suicidal behaviour [59]. Third, selection bias might have occurred because the parents of clinical cases were used to talking about their children's lives and motivated to change the health care system. However, we noticed that the parents of "out of the blue" cases were also motivated to participate because they had many unanswered questions about their children's death by suicide. Fourth, as in other studies, youths with a migrant background were under-represented. Approaches specifically targeting migrant communities may be used to overcome this issue in the future.

## Conclusions and recommendations for suicide prevention

This study showed that, as expected, a variety of mostly complex clusters of problems played a role in youth suicides. The study has the following implications for practice. A structural implementation of the PA procedure in the Netherlands should be prioritized. This requires the following concrete building blocks: facilitation of bereavement support, uniform implementation guidelines, a more automated form of data collection, real-time surveillance data, and a national database. By doing so, a learning system would be implemented to continuously monitor, evaluate, and provide postvention after each youth suicide and thereby improve suicide prevention strategies. Second, the two dominant patterns that were found in early adolescence highlight the importance of multimodal preventive programmes aimed at the early screening and detection of mental health problems and suicidal behaviours among students in secondary education [60]. In addition, promoting mental health literacy and skill training is needed to teach students to better cope with personal difficulties and performance pressure. Third, this study makes clear that more regional expertise centres and research are needed to

understand and treat suicidal behaviour in adolescents with multiple psychiatric problems (e.g. autism, eating disorders, and depression) [61]. Fourth, there is a need for monitoring adherence to the World Health Organization's suicide contagion media guidelines on popular social media platforms, such as Instagram. In particular, the effects of communicating protective coping strategies online and providing resources for help should be examined [38]. In addition, parents and professionals should monitor the social media activities of adolescents more actively to prevent their engagement in unhealthy social networks. Finally, the prevention of hanging and train suicides remains a major challenge to be addressed in youth suicides in the Netherlands, and stakeholders, such as the National Railway Services, are needed to create more safety in this respect.

## Supporting information

**S1 File. COREQ checklist.** This is a 32-time checklist of the consolidated criteria for reporting qualitative studies (COREQ).
(DOCX)

**S2 File. Interview guide.** The English version of the interview guide that was used in the psychological autopsy study.
(DOCX)

**S3 File. Interview guide.** The Dutch version of the interview guide that was used in the psychological autopsy study.
(DOCX)

## Acknowledgments

We thank the parents who participated in this study for their important contribution to suicide prevention by sharing their stories about the life course and painful loss of their children by suicide. Anne Roos, Alice Schutte and Henk van der Beld are thanked for their great endeavour as senior interviewers and recruitment coordinator, and interns Pommeline van der Post, Laura Wienen and Thara Boot for their important recruitment activities and coding work. Finally, we thank the members of the advisory committee, including Prof. Ella Arensman and Prof. Gwendolyn Portzky amongst others, and the chairman Prof. Ad Kerkhof for their constructive and helpful comments during the research trajectory.

## Author Contributions

**Conceptualization:** Saskia Mérelle, Diana Van Bergen, Sanne Rasing, Lieke van Domburgh, Maaike Nauta, Onno Sijperda, Wico Mulder, Renske Gilissen, Gerdien Franx, Daan Creemers, Arne Popma.

**Data curation:** Milou Looijmans, Elias Balt.

**Formal analysis:** Saskia Mérelle, Diana Van Bergen, Milou Looijmans, Elias Balt, Sanne Rasing.

**Funding acquisition:** Renske Gilissen, Gerdien Franx.

**Methodology:** Saskia Mérelle, Diana Van Bergen, Sanne Rasing, Lieke van Domburgh, Maaike Nauta, Onno Sijperda, Daan Creemers, Arne Popma.

**Project administration:** Milou Looijmans, Elias Balt.

**Software:** Milou Looijmans, Elias Balt.

**Supervision:** Saskia Mérelle, Diana Van Bergen, Arne Popma.

**Writing – original draft:** Saskia Mérelle, Diana Van Bergen.

**Writing – review & editing:** Milou Looijmans, Elias Balt, Sanne Rasing, Lieke van Domburgh, Maaike Nauta, Wico Mulder, Renske Gilissen, Gerdien Franx, Daan Creemers, Arne Popma.

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
