## [Decision Letter · Decision Letter 0]

10 Aug 2020

A multi-method psychological autopsy study on youth suicides in the Netherlands in 2017: feasibility, main outcomes, and recommendations

PONE-D-20-12782

Dear Dr. Merelle,

We’re pleased to inform you that your manuscript has been judged scientifically suitable for publication and will be formally accepted for publication once it meets all outstanding technical requirements.

Kind regards,

Geilson Lima Santana, M.D., Ph.D.

Academic Editor

PLOS ONE

Journal Requirements:

1. Please include a copy of the topic/interview guide used in the study, in both the original language and English, as Supporting Information, or include a citation if it has been published previously.

Reviewers' comments:

Reviewer's Responses to Questions

**Comments to the Author**

1. Is the manuscript technically sound, and do the data support the conclusions?

Reviewer #1: Yes

Reviewer #2: Yes

2. Has the statistical analysis been performed appropriately and rigorously? 

Reviewer #1: N/A

Reviewer #2: Yes

3. Have the authors made all data underlying the findings in their manuscript fully available?

Reviewer #1: Yes

Reviewer #2: No

4. Is the manuscript presented in an intelligible fashion and written in standard English?

Reviewer #1: Yes

Reviewer #2: Yes

5. Review Comments to the Author

Reviewer #1: This article is very interesting and represents a valuable contribution to knowledge about adolescent suicide. It is worth mentioning that on a personal basis, the observations made in the article coincide with my own clinical observation working with patients, after a suicidal attempt, in Chile. This relieves the transversality of the observations made.

Reviewer #2: The article explores an original topic using a qualitative as well as a quantitative methods. Knowledge and prevention of suicide in Young people are a challenge of mental health. No additional changes are necessary and I think that teh manuscript could be accepted without modifications.

6. PLOS authors have the option to publish the peer review history of their article (what does this mean?). If published, this will include your full peer review and any attached files.

Reviewer #1: **Yes: **Maria de la Paz Maino

Reviewer #2: **Yes: **Loas Gwenolé

---

## [Editor Report · Acceptance letter]

17 Aug 2020

PONE-D-20-12782 

A multi-method psychological autopsy study on youth suicides in the Netherlands in 2017: feasibility, main outcomes, and recommendations 

Dear Dr. Mérelle:

I'm pleased to inform you that your manuscript has been deemed suitable for publication in PLOS ONE. Congratulations! Your manuscript is now with our production department. 

Kind regards, 

on behalf of

Dr. Geilson Lima Santana 

Academic Editor

PLOS ONE